# Machine learning for predicting antimicrobial resistance in critical and high-priority pathogens: A systematic review considering antimicrobial susceptibility tests in real-world healthcare settings

**Carlos M. Ardila**[1,2]*, **Daniel González-Arroyave**[3], **Sergio Tobón**[2]

**1** Basic Sciences Department, Biomedical Stomatology Research Group, Faculty of Dentistry, Universidad de Antioquia U de A, Medellín Colombia, **2** Postdoctoral Program, CIFE University Center, Cuernavaca, México, **3** Department of Surgery, Universidad Pontificia Bolivariana, Medellín, Colombia

* martin.ardila@udea.edu.co

## Abstract

### Background

Antimicrobial resistance (AMR) poses a worldwide health threat; quick and accurate identification of AMR enhances patient outcomes and reduces inappropriate antibiotic usage. The objective of this systematic review is to evaluate the efficacy of machine learning (ML) approaches in predicting AMR in critical and high-priority pathogens (CHPP), considering antimicrobial susceptibility tests in real-world healthcare settings.

### Methods

The search methodology encompassed the examination of several databases, such as PubMed/MEDLINE, EMBASE, Web of Science, SCOPUS, and SCIELO. An extensive electronic database search was conducted from the inception of these databases until November 2024.

### Results

After completing the final step of the eligibility assessment, the systematic review ultimately included 21 papers. All included studies were cohort observational studies assessing 688,107 patients and 1,710,867 antimicrobial susceptibility tests. GBDT, Random Forest, and XGBoost were the top-performing ML models for predicting antibiotic resistance in CHPP infections. GBDT exhibited the highest AuROC values compared to Logistic Regression (LR), with a mean value of 0.80 (range 0.77–0.90) and 0.68 (range 0.50–0.83), respectively. Similarly, Random Forest generally showed better AuROC values compared to LR (mean value 0.75, range 0.58–0.98 versus mean value 0.71, range 0.61–0.83). However, some predictors selected by these algorithms align with those suggested by LR.

**Data availability statement:** All relevant data are within the paper and its Supporting Information files.

**Funding:** The author(s) received no specific funding for this work.

**Competing interests:** The authors have declared that no competing interests exist.

## Conclusions

ML displays potential as a technology for predicting AMR, incorporating antimicrobial susceptibility tests in CHPP in real-world healthcare settings. However, limitations such as retrospective methodology for model development, nonstandard data processing, and lack of validation in randomized controlled trials must be considered before applying these models in clinical practice.

## Introduction

Antimicrobial resistance (AMR) refers to bacteria's capacity to resist antimicrobial managements, notably antibiotics. Infections resulting from antibiotic-resistant bacteria present a significant challenge to contemporary healthcare [1]. Therefore, AMR poses a significant public health threat, with the projected number of deaths from bacterial infections expected to reach nearly 10 million per year by 2050 on a global scale [2]. A thorough study assessing the effects of antibiotic-resistant bacteria (ARBs) on human health found that in 2019, ARBs were directly linked to 1.27 million deaths and contributed to 4.95 million more fatalities worldwide [3]. ARBs represent a major contributor to mortality in resource-limited settings [3,4]. Many of these deaths result from infections caused by pathogens such as *Escherichia coli, Staphylococcus aureus, Klebsiella pneumoniae, Acinetobacter baumannii,* and *Pseudomonas aeruginosa*-organisms categorized as critical or high-priority by the World Health Organization (WHO) [1]. In 2019, methicillin-resistant *S. aureus* (MRSA) alone accounted for over 100,000 deaths linked to antimicrobial resistance (AMR) globally, while six additional pathogen-drug combinations, including multidrug-resistant *E. coli,* fluoroquinolone-resistant *E. coli,* carbapenem-resistant *A. baumannii* and *K. pneumoniae,* and third-generation cephalosporin-resistant *K. pneumoniae,* each caused between 50,000 and 100,000 fatalities across the globe. Moreover, a recent analysis of the antibiotic development pipeline highlighted the progress of 50 new drugs, but only 12 have demonstrated effectiveness against specific high-priority Gram-negative bacteria [5,6].

Timely administration of effective antimicrobials has been shown to drastically improve survival rates. In cases of bacteremia, failing to provide appropriate antibiotics within 24 hours can double the risk of mortality. However, on a global scale, only around half of all antibiotic prescriptions are accurate, underscoring the urgent need for rapid and reliable point-of-care diagnostic tools to tackle this issue [1,7].

Traditional culture-based techniques for pathogen detection remain inadequate for the demands of modern clinical settings. These methods typically require 24–48 hours to identify culturable bacteria associated with infections. An additional 2–4 hours is often needed for pathogen identification, and if antimicrobial resistance is suspected, antibiotic susceptibility testing (AST) can take an extra 18–24 hours. As a result, the total time from sample collection to receiving actionable AST results can extend to 2–4 days in practice [1,8].

Innovative micro- and nanotechnology approaches for bacterial identification and AST are emerging to address these limitations. These include phenotypic techniques like microfluidic bacterial cultures and molecular methods such as multiplex PCR, hybridization probes, synthetic biology, nanoparticles, and mass spectrometry. Despite advancements in PCR and MALDI-TOF mass spectrometry for bacterial identification in positive cultures, these technologies face notable challenges. PCR depends on predefined targets, while MALDI-TOF remains prohibitively expensive for widespread use [1,8,9] Moreover, these methods, which typically focus on detecting specific resistance genes, fail to accurately predict phenotypic antimicrobial

susceptibility. This limitation arises because resistance phenotypes often result from a complex interaction of resistance genes, regulatory mechanisms, and mutations [9].

Determining the appropriate empiric antibiotic for prescription remains challenging due to the limited availability of direct comparative trials, especially considering the myriads of patient-specific factors and evolving institutional AMR trends [10]. Clinical decisions regarding the choice of empiric antibiotic largely hinge on limited clinical evidence, typically centered on average treatment effects. This approach may result in suboptimal and undesirable outcomes such as inadequate early clinical response, prolonged hospitalization, and ultimately, heightened resistance [10,11].

Some investigations have focused on devising prediction algorithms tailored for personalized antimicrobial therapy to tackle challenges in infectious diseases [12–15]. When developing a clinical prediction model, it is crucial to define the prediction problem using readily available data in a scenario that closely mirrors real-world cases. Consequently, research endeavors should construct prediction models utilizing clinically significant variables, encompassing all predictors delineated by clinical guidelines, and draw conclusions that align with practical clinical considerations [10]. However, many studies have constructed prediction models based solely on key variables considered clinically significant, thereby overlooking certain predictors outlined in clinical guidelines—reflecting actual clinical decisions. Local antibiogram data, for instance, often goes unconsidered [16]. Consequently, these models have failed to yield a framework suitable for application in hospitalized patients, and their translation into clinical practice has been limited [10].

Various studies have employed different machine learning (ML) techniques to forecast AMR profiles for diverse bacterial species and drug combinations [16–18]. ML techniques have been employed to forecast antibiotic resistance in bloodstream infections, urinary tract infections, and genetic data of pathogens [18,19]. While these methodologies offer the potential for uncovering novel clinical insights, their widespread adoption remains limited due to challenges in integrating them into clinical workflows, issues surrounding interpretability, and a dearth of evidence showcasing their applicability and efficacy in real-world clinical environments [19].

Considering that various recent studies have identified CHPP as the predominant pathogens and the least susceptible species to antimicrobials in hospital-acquired infections [10,12,14,15], the emergence of multidrug-resistant bacteria presents a significant challenge in healthcare. This underscores the urgent need for innovative approaches to analyze and intervene in antimicrobial resistance. ML offers a powerful toolkit for dissecting the intricate web of factors influencing multidrug resistance [10,14,15]. By leveraging large-scale datasets encompassing clinical, microbiological, and antimicrobial susceptibility tests, and epidemiological variables, ML models can discern subtle patterns and relationships that traditional statistical methods may overlook. These models hold the potential to identify novel risk factors, predict patient outcomes, and inform personalized treatment strategies tailored to combat multidrug resistance effectively [10,15].

Despite the increasing literature on ML applications in AMR, there is a lack of comprehensive synthesis of existing evidence, particularly regarding antimicrobial susceptibility tests for CHPP in real-world settings. This systematic review provides a structured approach to collating, analyzing, and synthesizing findings from disparate studies, offering insights that transcend individual investigations. By systematically evaluating the strengths and limitations of ML models in predicting and quantifying CHPP antimicrobial resistance, we can identify gaps in knowledge, assess the methodological rigor of existing studies, and delineate avenues for future research. The aim of this systematic review is to assess the effectiveness of machine learning methods in forecasting antimicrobial resistance in critical and high-priority

pathogens, with a focus on antimicrobial susceptibility testing within practical healthcare environments.

## Materials and methods

### Protocol and registration

The systematic review utilized a search methodology in accordance with PRISMA (Preferred Reporting Items for Systematic Reviews and Meta-analyses) guidelines [20] (S1 File). The systematic review protocol was officially registered on PROSPERO and can be identified by the code CRD42024527410.

### Eligibility criteria

This systematic review was conducted based on a research question designed using the Population, Intervention, Comparison, and Outcomes (PICO) framework:

P: Hospitalized patients subjected to culture and antibiotic susceptibility testing.
I: Application of machine learning techniques.
C: Alternative conventional prediction methods.
O: Prediction of antimicrobial resistance in CHPP using predictive performance metrics.

This review encompassed studies assessing the efficacy of ML in predicting antimicrobial resistance in CHPP, employing data obtained from hospital information systems and antimicrobial susceptibility tests. The exclusion criteria encompassed case reports and case series, as well as in vitro and animal studies. Additionally, abstracts, conference proceedings, brief communications, reviews, and studies lacking essential details regarding ML methods and predictive performance metrics were excluded.

### Information sources

The search methodology encompassed the examination of several scientific databases, such as PubMed/MEDLINE, EMBASE, Web of Science, SCOPUS, and SCIELO, in addition to a review of gray literature sources through Google Scholar. A broad search of electronic databases was performed, covering all records from their inception up to November 2024, with no restrictions on language. Additionally, supplementary records were sourced by meticulously reviewing the reference lists and citations of all full-text articles deemed eligible for inclusion in the systematic review.

### Search strategy

The search strategy employed the following terms: *"antimicrobial resistance"* OR *"antibiotic resistance"* AND *"microbial"* OR *"bacterial"* AND *"Escherichia coli"* AND *"Staphylococcus aureus"* AND *"Klebsiella pneumoniae"* AND *"Acinetobacter baumannii"* AND *"Pseudomonas aeruginosa"* AND *"infection"* AND *"machine learning"* OR *"deep learning"* OR *"prediction model"* OR *"risk assessment"* OR *"risk prediction"*. Tailored syntax and operators were applied to each database to ensure accurate retrieval of articles matching the specified terms. Adjustments were made to align with the unique search functionalities and syntax rules of individual databases. Table 1 summarizes the search protocols used for each database along with the corresponding search terms.

### Study selection

Two authors independently reviewed titles and abstracts to ascertain eligibility, followed by a comprehensive analysis of full-text articles. The determination of eligibility through full-text

**Table 1. Search approaches for the designated databases using the provided terms.**

| Database | Search strategy |
|---|---|
| PubMed/MEDLINE | (("Antimicrobial resistance" OR "antibiotic resistance") AND ("microbial" OR "bacterial") AND *"Escherichia coli"* AND *"Staphylococcus aureus"* AND *"Klebsiella pneumoniae"* AND *"Acinetobacter baumannii"* AND *"Pseudomonas aeruginosa"* AND "infection" AND ("machine learning" OR "deep learning" OR "prediction model" OR "risk assessment" OR "risk prediction")) |
| Scopus | TITLE-ABS-KEY(("Antimicrobial resistance" OR "antibiotic resistance") AND ("microbial" OR "bacterial") AND *"Escherichia coli"* AND *"Staphylococcus aureus"* AND *"Klebsiella pneumoniae"* AND *"Acinetobacter baumannii"* AND *"Pseudomonas aeruginosa"* AND "infection" AND ("machine learning" OR "deep learning" OR "prediction model" OR "risk assessment" OR "risk prediction")) |
| Scielo | ("Antimicrobial resistance" OR "antibiotic resistance") AND ("microbial" OR "bacterial") AND *"Escherichia coli"* AND *"Staphylococcus aureus"* AND *"Klebsiella pneumoniae"* AND *"Acinetobacter baumannii"* AND *"Pseudomonas aeruginosa"* AND "infection" AND ("machine learning" OR "deep learning" OR "prediction model" OR "risk assessment" OR "risk prediction") |
| Embase | ('Antimicrobial resistance' OR 'antibiotic resistance') AND ('microbial' OR 'bacterial') AND *'Escherichia coli'* AND *'Staphylococcus aureus'* AND *'Klebsiella pneumoniae'* AND *'Acinetobacter baumannii'* AND *'Pseudomonas aeruginosa'* AND 'infection' AND ('machine learning' OR 'deep learning' OR 'prediction model' OR "risk assessment" OR "risk prediction") |
| Web of Science | TS = ("Antimicrobial resistance" OR "antibiotic resistance") AND TS = ("microbial" OR "bacterial") AND TS=*"Escherichia coli"* AND *"Staphylococcus aureus"* AND *"Klebsiella pneumoniae"* AND *"Acinetobacter baumannii"* AND *"Pseudomonas aeruginosa"* AND TS="infection" AND TS = ("machine learning" OR "deep learning" OR "prediction model" OR "risk assessment" OR "risk prediction") |
| Google Scholar | "Antimicrobial resistance" OR "antibiotic resistance" AND "microbial" OR "bacterial" AND *"Escherichia coli"* *"Staphylococcus aureus"* AND *"Klebsiella pneumoniae"* AND *"Acinetobacter baumannii"* AND *"Pseudomonas aeruginosa"* AND "infection" AND "machine learning" OR "deep learning" OR "prediction model" OR "risk assessment" OR "risk prediction" |

scrutiny was conducted independently and redundantly. Any discrepancies were resolved through discussion, and if persistent disagreements arose, a third author was consulted. Interobserver agreement, with a predefined threshold of > 90, was assessed using the Kappa statistical test to determine statistical significance.

## Data collection

Two authors independently collected data using customized and study-specific data extraction templates designed to ensure consistent and accurate data retrieval. These templates were developed based on the research objectives and included predefined categories for capturing critical aspects such as resistance profiles, input variables, machine learning methodologies, performance metrics, and patient cohort sizes used for model development and validation. Additionally, the templates accounted for recording publication-specific details, such as authorship and publication year. Following data extraction, a comparative analysis was performed to harmonize discrepancies and ensure data reliability.

## Assessment of bias risk and study quality in individual studies

To evaluate the risk of bias and the applicability of prediction model studies for systematic reviews, the PROBAST framework was utilized [21]. This tool involves the assessment of 20 signaling questions grouped into four key areas: participants, predictors, outcomes, and

analysis. For each included study, particular attention was given to the first three domains. A domain was categorized as having a "high risk" of bias if at least one signaling question was answered as "no" or "probably no" without sufficient justification. Conversely, a domain was marked as "unclear risk" when critical details were missing for specific signaling items but did not meet the criteria for classification as high risk.

## Summary measurements

Descriptive statistics were employed to summarize the data collected from the included studies, focusing on continuous outcomes such as mean differences, standard deviations, and ranges. To ensure a comprehensive approach, data analysis included assessing the distribution of variables and identifying patterns or trends across studies. If substantial homogeneity was observed among the studies, a meta-analysis was considered feasible. The decision to perform a meta-analysis was guided by evaluating statistical measures such as heterogeneity indices (e.g., $I^2$ statistic) and visual inspection of forest plots. In cases where meta-analysis was not viable, a qualitative synthesis was conducted to narratively summarize the findings and provide contextual insights.

Ethical approval is not applicable to this study.

# Results

## Study selection

After conducting the search as described, 843 studies were identified in electronic databases. After removing duplicates and applying eligibility criteria, 37 papers underwent a detailed full-text assessment. Exclusion during full-text review primarily occurred due to the omission of patient data obtained from hospital information systems and antimicrobial susceptibility tests of CHPP (S2 Table. List of excluded studies with reasons). After completing the final step of the eligibility assessment, the systematic review ultimately included 21 papers. Fig. 1 provides a comprehensive representation of the search flowchart.

## Characteristics of the studies

Table 2 presents the descriptive characteristics of the 21 studies incorporated in this systematic review [10,12,14,15,22–38]. The analysis encompasses papers published between 2000 [38] and 2023 [10,22,23,28]. Four studies were prospective cohorts [30,33,35,36], and the rest were retrospective, evaluating data from 688,107 patients. Most of these studies were carried out in the United States (43%) and conducted at a single hospital center. Prediction models were developed to forecast non-susceptible outcomes of CHPP using demographics, microbiology, antimicrobial susceptibility tests, prescribing data, routine clinical information, and patients' electronic medical records.

All the studies assessed 1,710,867 antimicrobial resistance test results as detailed in Table 3. The most common samples used in the revised studies were blood and urine. This table also displays the antibiotics that were tested. The antimicrobial resistance of CHPP was extensively studied using several antibiotics, including Aminoglycosides, Ciprofloxacin, Ampicillin, Ampicillin/sulbactam, Cefepime, Piperacillin/tazobactam, Ceftriaxone, Gentamicin, Imipenem, and Sulfamethoxazole/trimethoprim, among others. These antibiotics were commonly assessed to understand the patterns and trends of resistance in CHPP infections. The pathogens most frequently subjected to antimicrobial susceptibility testing were *E. coli, K. pneumoniae*, and *P. aeruginosa*, although other CHPPs were also extensively studied (Table 3).

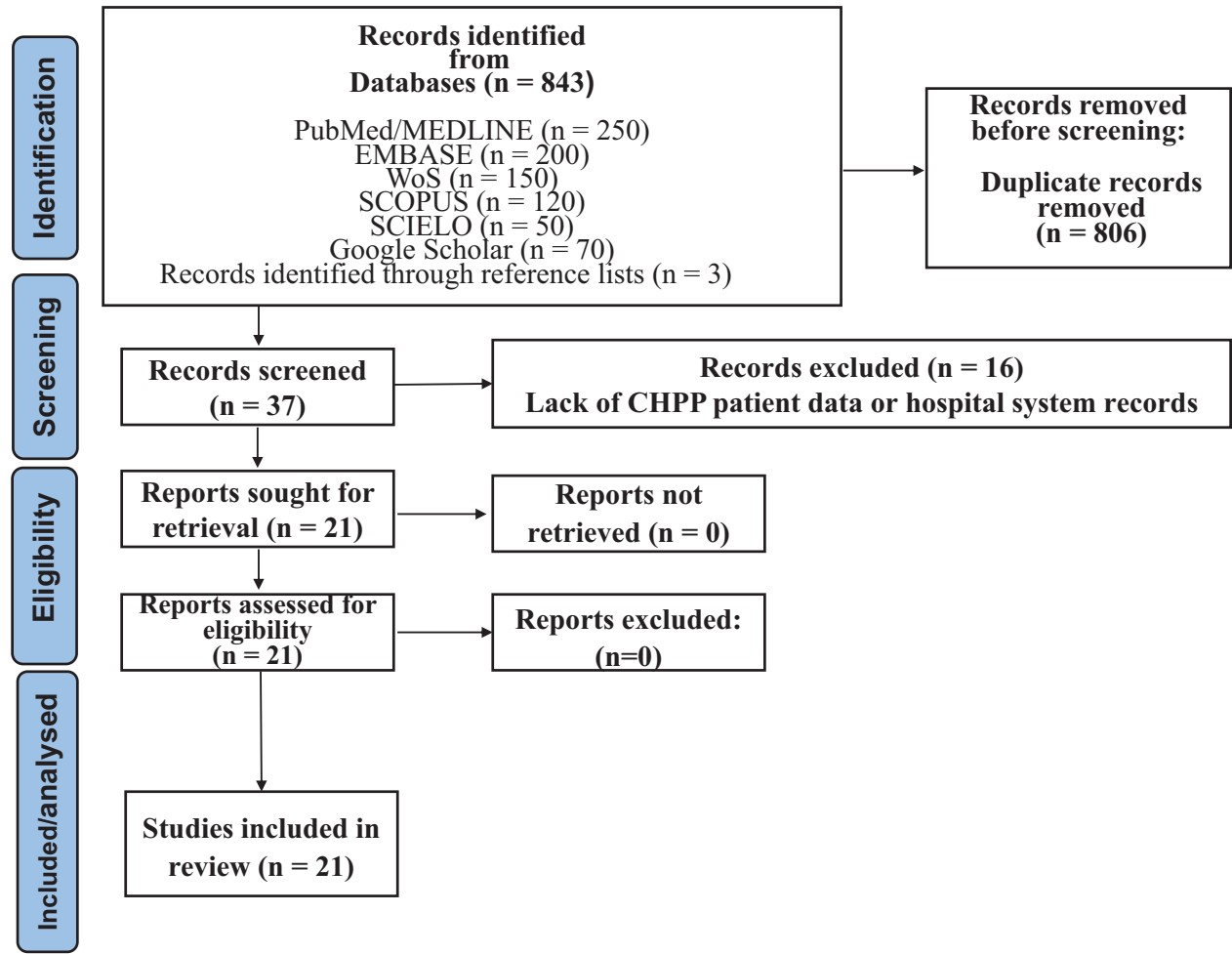

**Fig 1. Flowchart of the studies selection method.**

Table 4 presents other key findings and methodologies. The number of input features varied across different studies, ranging from 5 to 788 features. The most common ML models employed in the investigations were Random Forest, Gradient Boosting Decision trees (GBDT), XGBoost, Neural Networks, and Logistic Regression (LR). The area under the receiver operating characteristic curve (AuROC) was the most frequently used performance evaluation metric. Moreover, the most prevalent validation technique used was k-fold cross-validation.

All the included studies specified resistance patterns of CHPP. Various features were utilized as risk factors using hospitalized patients' electronic medical records. The most common predicted resistance pattern in these 21 studies was non-susceptibility in antibiotic tests. Most studies indicate that the incorporation of antimicrobial resistance testing data into ML models is essential for improving predictive performance, enhancing antibiotic stewardship, enabling personalized medicine, and providing valuable clinical decision support. These findings underscore the critical role of accurate resistance testing in addressing AMR effectively in real-world healthcare settings. Mintz et al. [23] highlighted the significance of AMR testing by identifying key variables such as previous resistance in the past 60 days and recent resistance to any antibiotic in hospital settings, underscoring the importance of incorporating such data

**Table 2. Overview of included studies.**

| Authors and publication year | Country | Study design | Number of Centers | Patients |
|---|---|---|---|---|
| Lee et al. 2023 [28] | South Korea | Retrospective cohort study | 1 | 550 |
| Tran Quoc et al. 2023 [22] | Vietnam | Retrospective cohort study | 2 | 1244 |
| Mintz et al. 2023 [23] | Israel | Retrospective cohort study | 1 | 5540 |
| Kim et al. 2023 [10] | South Korea | Retrospective cohort study | 1 | 10474 |
| Rich et al. 2022 [29] | USA | Retrospective cohort study | 2 | 9990 |
| Luterbach et al. 2022 [30] | USA | Prospective cohort study | 171 | 49 |
| Çağlayan et al. 2022 [31] | USA | Retrospective cohort study | 1 | 3958 |
| Weis et al. 2022 [24] | Switzerland | Retrospective cohort study | 4 | 303195 |
| Tzelves et al. 2022 [25] | Greece | Retrospective cohort study | 1 | 239 |
| Corbin et al. 2022 [26] | USA | Retrospective cohort study | 2 | 6920 |
| Lee et al. 2021 [32] | China | Retrospective cohort study | 3 | 5625 |
| Lewin-Epstein et al. 2021 [15] | Israel | Retrospective cohort study | 1 | Not reported |
| Moran et al. 2020 [27] | United Kingdom | Retrospective cohort study | 3 | 9352 |
| Kanjilal et al. 2020 [14] | USA | Retrospective cohort study | 2 | 10053 |
| Yelin et al. 2019 [12] | Israel | Retrospective cohort study | 1 | 315047 |
| Souza et al. 2019 [33] | Spain | Prospective cohort study | 1 | 448 |
| Goodman et al. 2019 [34] | USA | Retrospective cohort study | 1 | 194 |
| Goodman et al. 2019 [35] | USA | Prospective cohort study | 1 | 2165 |
| Hartvigsen et al. 2018 [36] | USA | Prospective cohort study | 1 | 1304 |
| Goodman et al. 2016 [37] | USA | Retrospective cohort study | 1 | 1288 |
| Shang et al. 2000 [38] | USA | Retrospective cohort study | 2 | 472 |

into ML models. Kim et al. [10] and Lewin-Epstein et al. [15] demonstrated that ML models, informed by AMR testing data, outperformed traditional methods in predicting antimicrobial susceptibility, as measured by AUROC. This emphasizes the critical role of accurate resistance testing data in training predictive models. Corbin et al. [26] and Kanjilal et al. [14] showcased the potential of ML in improving antibiotic stewardship by leveraging AMR testing results. These studies highlighted the importance of reducing unnecessary use of broad-spectrum antibiotics based on resistance patterns identified through testing. Yelin et al. [12] and Souza et al. [33] demonstrated the importance of incorporating AMR testing data into ML models for personalized medicine. These studies showed that combining patient demographics, clinical history, and resistance testing results enables tailored treatment recommendations, enhancing patient care. Lee et al. [28], Lee et al. [32], and Goodman et al. [37] illustrated the value of incorporating AMR testing into ML models as clinical decision support tools. These models aid in predicting resistance, adjusting empirical antibiotic treatment, and identifying patients at risk of resistant infections with high accuracy. Luterbach et al. [30] and Rich et al. [29] emphasized the importance of including AMR testing data in predictive models to improve accuracy. By integrating resistance testing results with clinical and bacterial variables, these models can better predict outcomes such as 30-day mortality and resistance development.

The AUROC values for ML prediction of antibiotic resistance in CHPP are presented in Table 5. Gradient Boosted Decision Trees (GBDT), Random Forest, and XGBoost consistently emerged as the top-performing models for predicting antibiotic resistance in CHPP infections. Among these, GBDT exhibited the highest AUROC values, with a mean of 0.80 (range: 0.77–0.90), outperforming Logistic Regression (LR), which had a mean AUROC of 0.68 (range: 0.50–0.83). Random Forest also demonstrated superior performance compared to LR, achieving a mean AUROC of 0.75 (range:

**Table 3. Summary of Antimicrobial Resistance Testing: Number of Test Results, Tested Antibiotics, and Studied Microorganisms.**

| Authors | Sample | Antimicrobial Susceptibility Test results | Antibiotics | Critical and High-Priority Pathogens Studied |
|---|---|---|---|---|
| Lee et al. [28] | Urine | 1100 | Ciprofloxacin Cefotaxime and ceftazidime alone and in combination with clavulanate | *Escherichia coli, and Klebsiella pneumoniae* |
| Tran Quoc et al. [22] | Blood Cerebrospinal fluid Tracheobronchial/ bronchoalveolar fluid Urine Skin/wound/ tissue specimens. Catheters Pleural Peritoneal fluid | 2719 | Aminoglycosides Carbapenem Fourth-generation cephalosporin Trimethoprim derivatives | *Escherichia coli, Staphylococcus aureus, Klebsiella pneumoniae, and Pseudomonas aeruginosa* |
| Mintz et al. [23] | Blood Urine Wound | 10053 | Ciprofloxacin | *Escherichia coli, Staphylococcus aureus, Klebsiella pneumoniae, and Pseudomonas aeruginosa* |
| Kim et al. [10] | Urine | 42156 | Ampicillin Ampicillin/ sulbactam Cefepime Ciprofloxacin Gentamicin Imipenem Piperacillin/ tazobactam Sulfamethoxazole/ trimethoprim | *Escherichia coli, Staphylococcus aureus, Klebsiella pneumoniae, Acinetobacter baumannii, and Pseudomonas aeruginosa* |
| Rich et al. [29] | Not reported | 9990 | Sulfamethoxazole/ trimethoprim Nitrofurantoin Ciprofloxacin | *Escherichia coli, Staphylococcus aureus, Klebsiella pneumoniae, and Pseudomonas aeruginosa* |
| Luterbach et al. [30] | Blood | 22 | Colistin Ceftazidime/ avibactam | *Klebsiella pneumoniae* |
| Çağlayan et al. [31] | Peri-rectal Nasal Blood Urine Wound | 16470 | Aminoglycosides Aztreonams Carbapenems Cephalosporins Fluoroquinolones Penicillin | *Escherichia coli, Staphylococcus aureus, and Klebsiella pneumoniae* |
| Weis et al. [24] | Blood Stool Genital Respiratory Deep tissues | 768300 | Ceftriaxone Ciprofloxacin Cefepime Piperacillin/ Tazobactam Tobramycin | *Escherichia coli, Staphylococcus aureus, and Klebsiella pneumoniae* |
| Tzelves et al. [25] | Blood Urine Pus | 5156 | 38 antibiotics | *Escherichia coli, Staphylococcus aureus, Klebsiella pneumoniae, Acinetobacter baumannii, and Pseudomonas aeruginosa* |
| Corbin et al. [26] | Blood Urine Cerebral spinal fluid | 8342 | Vancomycin Piperacillin/ tazobactam Cefepime Ceftriaxone Cefazolin Ciprofloxacin Ampicillin Meropenem | *Escherichia coli, Staphylococcus aureus, Klebsiella pneumoniae, and Pseudomonas aeruginosa* |
| Lee et al. [32] | Blood | 5625 | Amoxicillin/ clavulanate Piperacillin/ tazobactam Third-generation cephalosporin Fourth-generation cephalosporin Carbapenem Quinolones | *Escherichia coli, and Klebsiella pneumoniae* |

*(Continued)*

**Table 3.** (Continued)

| Authors | Sample | Antimicrobial Susceptibility Test results | Antibiotics | Critical and High-Priority Pathogens Studied |
|---|---|---|---|---|
| Lewin-Epsteinet al. [15] | Blood Urine | 16198 | Ceftazidime Gentamicin Imipenem Ofloxacin, Sulfamethoxazole/ trimethoprim | *Escherichia coli, Staphylococcus aureus, Klebsiella pneumoniae, and Pseudomonas aeruginosa* |
| Moran et al. [27] | Blood Urine | 15580 | Co-amoxiclav Piperacillin/ Tazobactam | *Escherichia coli, Klebsiella pneumoniae, and Pseudomonas aeruginosa* |
| Kanjilal et al. [14] | Urine | 11865 | Ciprofloxacin Levofloxacin Nitrofurantoin Trimethoprim/ Sulfamethoxazole | *Escherichia coli, Staphylococcus aureus, and Klebsiella pneumoniae* |
| Yelin et al. [12] | Urine | 711099 | Trimethoprim-Sulfa Ciprofloxacin Nitrofurantoin Amoxicillin-CA Cefuroxime axetil Cephalexin | *Escherichia coli, and Klebsiella pneumoniae* |
| Souza et al. [33] | Blood | 132 | Combinations of cephalosporins with clavulanic acid | *Escherichia coli, Klebsiella pneumoniae, and Pseudomonas aeruginosa* |
| Goodman et al. [34] | Blood | 1288 | Ceftriaxone Combinations of cephalosporins with clavulanic acid | *Escherichia coli, and Klebsiella pneumoniae* |
| Goodman et al. [35] | Peri-rectal | 2878 | Ertapenem, Meropenem Imipenem | *Escherichia coli, Staphylococcus aureus, Klebsiella pneumoniae, and Pseudomonas aeruginosa* |
| Hartvigsen et al. [36] | Blood Urine | 80293 | Methicillin | *Pseudomonas aeruginosa* |
| Goodman et al. [37] | Blood | 1288 | Ceftriaxone Extended-spectrum penicillin Third- and fourth generation cephalosporins Aztreonam Carbapenems Aminoglycosides Fluoroquinolones | *Escherichia coli, and Klebsiella pneumoniae* |
| Shang et al. [38] | Blood Urine Respiratory tract Wound Feces | 313 | Aminoglycosides Cephalosporins Vancomycin Quinolones Penicillin | *Pseudomonas aeruginosa* |

0.58–0.98), while LR had a slightly broader range. Interestingly, many of the predictors identified by these advanced models aligned with those derived from LR, showcasing the robustness of these approaches. Collectively, these machine learning models leveraged antimicrobial susceptibility test data from real-world healthcare environments to accurately predict resistance patterns.

Fig 2 visually complements these findings, summarizing the reported AUROC values across different studies and ML models. The heatmap highlights the variability in performance across studies, with darker shades representing higher AUROC values (closer to 1.0), indicative of better model performance. The vertical axis organizes studies, while the horizontal axis categorizes machine learning models, offering a clear comparative framework. Notably, GBDT, XGBoost, and Random Forest consistently demonstrated high AUROC values across multiple studies, reinforcing their reliability in this domain. Meanwhile, Logistic Regression, despite exhibiting lower overall performance, occasionally approached comparable predictive accuracy in specific datasets.

These results underscore the potential of ensemble-based models like GBDT and Random Forest in advancing the predictive accuracy of antibiotic resistance in CHPP, particularly when integrated with robust datasets from clinical environments.

Table 4. Key Findings and Methodologies.

| Authors/ Publication year | Quantity of input characteristics | Machine Learning model | Assessment of performance |
|---|---|---|---|
| Lee et al. 2023 [28] | 39 | GBDT<br>LR | Sensitivity<br>Specificity<br>Precision<br>AuROC<br>k-fold<br>cross-validation |
| Tran Quoc et al. 2023 [22] | 22 | LR<br>AdaBoost.<br>Random Forest<br>XGBoost<br>LightGBDT | Sensitivity<br>Specificity<br>Precision<br>Accuracy<br>AuROC<br>normMCC<br>PRC<br>F1-score |
| Mintz et al. 2023 [23] | 73 | LASSO LR.<br>Random Forest<br>GBDT<br>Neural networks | Sensitivity<br>Mean observed probability.<br>AuROC<br>k-fold<br>cross-validation |
| Kim et al. 2023 [10] | 140 | LASSO LR<br>XGBoost<br>Random Forest<br>Stacked ensemble method | AuROC<br>AuPCR<br>k-fold<br>cross-validation |
| Rich et al. 2022 [29] | 41 | Boosted LR<br>Random Forest<br>Decision tree | Sensitivity<br>Specificity<br>AuROC<br>Boot- strap validation |
| Luterbach et al. 2022 [30] | 34 | Random Forest | nCV |
| Çağlayan et al. 2022 [31] | 11 | LASSO LR<br>Random Forest<br>XGBoost | Sensitivity Specificity<br>AuROC<br>k-fold<br>cross-validation |
| Weis et al. 2022 [24] | 30 | LR<br>LightGBDT<br>MLP | AuROC<br>AuPCR<br>k-fold<br>cross-validation |
| Tzelves et al. 2022 [25] | 55 | WEKA-Data<br>LR | AuROC<br>AuPCR<br>k-fold<br>cross-validation |
| Corbin et al. 2022 [26] | 788 | LASSO LR<br>Ridge LR<br>Random Forest<br>GBM | AuROC<br>k-fold<br>cross-validation |
| Lee et al. 2021 [32] | 136 | LR<br>Neural network | Sensitivity Specificity<br>AuROC<br>PPV<br>NPV<br>Accuracy<br>F1-score |
| Lewin-Epsteinet al. 2021 [15] | 448 | LASSO LR<br>GBDT<br>Neural network<br>Ensemble that combined all 3 algorithms | AuROC<br>k-fold<br>cross-validation |

*(Continued)*

**Table 4.** (Continued)

| Authors/ Publication year | Quantity of input characteristics | Machine Learning model | Assessment of performance |
|---|---|---|---|
| Moran et al. 2020 [27] | 5 | XGBoost-GBDT<br>LR | AuROC |
| Kanjilal et al. 2020 [14] | 10 | LR<br>Decision tree<br>Random Forest | AuROC |
| Yelin et al. 2019 [12] | 18 | CML<br>UCML<br>RP<br>RD | AuROC |
| Souza et al. 2019 [33] | 5 | Decision tree | Sensitivity Specificity<br>AuROC<br>PPV<br>NPV |
| Goodman et al. 2019 [34] | 14 | Decision tree<br>LR | Sensitivity Specificity<br>AuROC<br>PPV<br>NPV<br>k-fold<br>cross-validation |
| Goodman et al. 2019 [35] | 3 | Decision tree | Sensitivity Specificity<br>AuROC<br>PPV<br>NPV<br>k-fold<br>cross-validation |
| Hartvigsen et al. 2018 [36] | 84 | LR<br>Random Forest<br>SVM | AuROC<br>Accuracy<br>Precision<br>Recall<br>F1-score |
| Goodman et al. 2016 [37] | 5 | LASSO LR<br>DT | AuROC<br>k-fold<br>cross-validation |
| Shang et al. 2000 [38] | 38 | LR<br>Neural Network | AuROC<br>k-fold<br>cross-validation |

Abbreviations: GBDT, gradient-boosted decision trees; LR, logistic regression; MLP, multi-layer perceptron; XGBoost, eXtreme Gradient Boosting; WEKA, data mining software in Java Workbench; SVC, support vector classification; SVM, support vector machine; SMO, sequential minimal optimization; kNN, k-nearest neighbors; RIPPER, repeated incremental pruning to produce error reduction; MLP, multilayer perceptron; CML, constrained machine learning model; UCML, unconstrained machine learning model; RP, random permutation model; RD, random dice model; AuROC, the area under the receiver operating characteristic curve; F1-score, the harmonic mean of precision and recall; normMCC, normalized Matthew Correlation Coefficient; PRC, precision-recall curve; AdaBoost, adaptive boosting decision trees; AUPRC, area under the precision- recall curve; nCV, nested cross-validation; PPV, positive predict value; NPV, negative predict value.

## Assessment of bias risk

Since the PROBAST tool suggests that "model development and validation studies pose a higher risk of bias when participant data come from existing sources like cohort studies or routine care registries," and if an "evaluation is deemed high for at least one domain, it should be considered to have "high risk of bias" or "high concern" regarding applicability [21]. Therefore, most studies included in this systematic review were considered to have a high risk

**Table 5. Comparison of AuROC values of different machine learning models.**

| Study | DT | GBDT | Random Forest | XGBoost | AdaBoost | Neural network | WEKA | LR |
|---|---|---|---|---|---|---|---|---|
| Lee et al. [28] | ----- | 0.83 | ------ | ------ | ------ | ------ | ------ | 0.50 |
| Tran Quoc et al. [22] | ----- | 0.99 | 0.98 | 0.99 | 0.95 | ----- | ------ | 0.83 |
| Mintz et al. [23] | ----- | ------ | 0.72 | 0.73 | ----- | 0.72 | ------ | 0.73 |
| Kim et al. [10] | ----- | ------ | 0.76 | 0.75 | ----- | ----- | ------ | 0.73 |
| Rich et al. [29] | 0.59 | ------ | 0.58 | ----- | ----- | ----- | ------ | 0.61 |
| Luterbach et al. [30] | ----- | ------ | 0.71 | ----- | ----- | ----- | ------ | ------ |
| Çağlayan et al. [31] | ------ | ------ | 0.80 | 0.77 | ----- | ----- | ------ | 0.73 |
| Weis et al. [24] | ------ | 0.74 | ------ | ----- | ----- | 0.68 | ------ | 0.70 |
| Tzelves et al. [25] | ------ | ------ | ------ | ------ | ------ | ------ | 0.87 | 0.77 |
| Corbin et al. [26] | ------ | 0.73 | 0.72 | ------ | ------ | ------ | ------ | 0.64 |
| Lee et al. 2021 [32] | ------ | ------ | ------ | ------ | ------ | 0.76 | ------ | 0.67 |
| Lewin-Epstein et al. [15] | ------ | ------ | ------ | 0.82 | ------ | 0.80 | ------ | 0.82 |
| Moran et al. [27] | ------ | 0.70 | ------ | ------ | ------ | ------ | ------ | 0.67 |
| Kanjilal et al. [14] | ------ | ------- | Poor validation | ------ | ------ | ------ | ------ | 0.64 |
| Yelin et al. [12] | ------ | 0.80 | ------ | ------ | ------ | ------ | ------ | 0.77 |
| Souza et al. [33] | 0.70 | ------ | ------ | ------ | ------ | ------ | ------ | ------ |
| Goodman et al. [34] | 0.77 | ------ | ------ | ------ | ------ | ------ | ------ | 0.87 |
| Goodman et al. [35] | 0.57 | ------ | ------ | ------ | ------ | ------ | ------ | ------ |
| Hartvigsen et al. 2018 [36] | ------ | ------ | 0.76 | ------ | ------ | ------ | ------ | 0.70 |
| Goodman et al. 2016 [37] | 0.78 | ------ | ------ | ------ | ------ | ------ | ------ | 0.78 |
| Shang et al. 2000 [38] | ------ | ------ | ------ | ------ | ------ | 0.93 | ------ | 0.87 |

Abbreviations: DT, decision tree; GBDT, gradient-boosted decision trees; XGBoost, eXtreme Gradient Boosting; AdaBoost, adaptive boosting; WEKA, data mining software in Java Workbench; LR, logistic regression.

of bias due to the retrospective nature of the 17 studied cohorts (Table 6). The remaining four prospective cohort studies were rated as low risk in terms of participant domain. However, in three of these studies, essential information for at least one item was missing [30,35,36], leading to classification as having a risk of bias according to the tool used.

## Discussion

A systematic review of ML prediction for CHPP antimicrobial resistance in real-world settings was conducted. While LR was commonly used for prediction, gradient-boosted decision trees were also frequently employed. However, GBDT exhibited the highest AuROC values compared to LR. Additional algorithms included Random Forest, XGBoost, and Neural Networks, among others. Certainly, all ML models forecasted the resistance patterns of CHPP against various antibiotics by utilizing data from real healthcare environments' antimicrobial susceptibility tests.

This is the first systematic review that assesses the validation of ML models for predicting AMR using antimicrobial susceptibility tests of CHPP in real-world studies. This recommendation was made by Tang et al. in a previous systematic review that studied ML as a potential technology for AMR prediction [39]. Concerningly, in Tang et al.'s review, about half of the studies examining ML predictions did not specify resistance patterns, whereas, in our review, all studies provided details on AMR patterns. Our study's inclusion of AMR patterns is crucial for improving the robustness and applicability of ML models in real-world settings. By providing details on resistance patterns, our review offers valuable insights into the effectiveness of these models for guiding antimicrobial therapy.

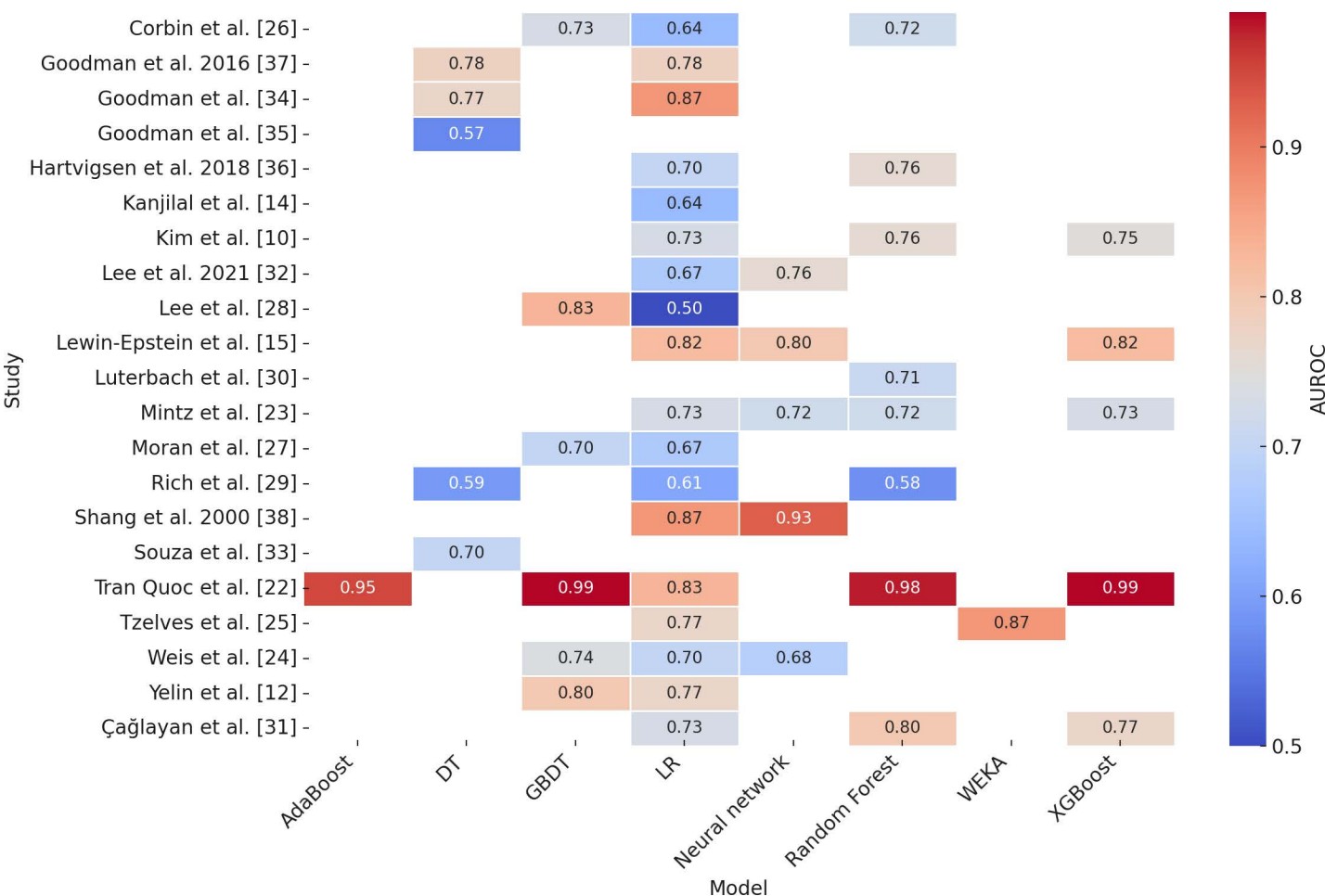

**Fig 2. Heatmap of Area Under the Receiver Operating Characteristic Curve (AUROC) values for various machine learning models across studies.**

The importance of including information related to AMR testing obtained in hospital settings has also been highlighted by various studies. Excluding antimicrobial susceptibility test information led to AuROC values ranging from 0.73 to 0.79.

However, including bacterial species resulted in even higher AuROC scores, ranging from 0.8 to 0.88 [15]. In Kim et al.'s study [10], the models incorporated a daily calculation function of the antibiotic non-susceptibility rate to common causative infections. This function was designed to incorporate both resistance and intermediate susceptible data based on local outcomes within 90 days of the index date. It enhanced the models' relevance by offering timely updates on the institution's non-susceptibility outcomes during the observation window periods of the cohort. By utilizing antimicrobial susceptibility tests and data from the electronic health record, a decision algorithm managed to reduce the prescription of second-line drugs by 67% and unnecessary antibiotic treatment by 18% compared to physicians [14]. By employing repeated cultures from the same patient stored in a database, it became feasible to identify and define a personalized aspect of memory-like correlations of resistance lasting for several months, or even years. These enduring connections might suggest recurrent infections with the same strain or correlations with other patient-specific factors. In both scenarios, it was shown that they contribute to the predictability of resistance [12]. Optimization simulations

**Table 6. Evaluation of risk bias [21].**

| Study | Risk of bias | | | | Applicability | Overall | | | |
|---|---|---|---|---|---|---|---|---|---|
| | Participants | Predictors | Outcome | Analysis | Participant | Predictor | Outcome | Risk of bias | Applicability |
| Lee et al. [28] | – | – | – | – | + | + | + | – | + |
| Tran Quoc et al. [22] | – | + | + | + | + | + | + | – | + |
| Mintz et al. [23] | – | + | + | + | + | + | + | – | + |
| Kim et al. [10] | – | + | + | + | + | + | + | – | + |
| Rich et al. [29] | – | + | + | + | + | + | + | – | + |
| Luterbach et al. [30] | + | + | + | – | + | + | + | – | + |
| Çağlayan et al. [31] | – | + | + | + | + | + | + | – | + |
| Weis et al. [24] | – | + | + | + | + | + | + | – | + |
| Tzelves et al. [25] | – | + | + | + | + | + | + | – | + |
| Corbin et al. [26] | – | + | + | + | + | + | + | – | + |
| Lee et al. 2021 [32] | – | + | + | + | + | + | + | – | + |
| Lewin-Epsteinet al. [15] | – | + | + | + | + | + | + | – | + |
| Moran et al. [27] | – | – | – | ? | + | + | + | – | + |
| Kanjilal et al. [14] | – | + | + | + | + | ? | + | – | ? |
| Yelin et al. [12] | – | + | + | + | + | + | + | – | + |
| Souza et al. [33] | + | + | + | + | + | + | + | + | + |
| Goodman et al. [34] | – | + | + | + | + | + | + | – | + |
| Goodman et al. [35] | + | + | + | – | + | + | + | – | + |
| Hartvigsen et al. 2018 [36] | + | + | ? | ? | + | + | + | ? | + |
| Goodman et al. 2016 [37] | – | + | + | + | + | + | + | – | + |
| Shang et al. 2000 [38] | – | – | – | – | + | + | + | – | + |

Abbreviations: +, low risk; –, high risk; ?, unclear risk.

demonstrate that, despite modest AUROCs, antibiotic selection guided by individualized antibiograms can either match or surpass clinician performance. Moreover, antibiotic selection based on individualized antibiograms led to coverage rates comparable to those seen in real-world scenarios, using less broad-spectrum antibiotics [26]. This underscores an ongoing and crucial antibiotic stewardship challenge.

The AuROC has long been a standard measure in assessing model performance. This parameter was extracted as the main performance metric in this systematic review and in a previous systematic review and meta-analysis evaluating antimicrobial resistance. However, the previous review did not consider antimicrobial susceptibility testing or real-world settings in all the studies analyzed [39]. Interestingly, the range of this value in the Logistic Regression results of the two studies is similar (0.58–0.89 versus 0.50–0.83), but it differs for ML results (0.48–0.92 versus 0.77–0.90 for GBDT in our study) and is like that of Random Forest analyzed in our study (0.48–0.92 versus 0.58–0.98). Comparing the results is obviously very difficult considering that the selection criteria of the two reviews are different, but it could be speculated that the inclusion of antimicrobial resistance tests in real-life contexts could make a difference and that, furthermore, the models behave differently depending on the input variables. In this context, it has been noted that although there are limitations in comparing results across different settings, the models developed by Mintz et al. [23] demonstrate high predictive performance compared to earlier studies [12,40]. Notably, these models performed well on a highly diverse dataset (with an AuROC of 0.73), encompassing various bacterial species, sample sources, and multiple hospital departments [23]. Feretzakis et al. [40]

predicted antibiotic resistance using data from a single internal medicine department, based on the sample's Gram stain result, achieving an AuROC of 0.72, while Yelin et al. [12] focused on predicting antibiotic resistance solely in outpatients, using urine samples and restricted to three bacterial species, with an AuROC of 0.83.

Usually, predictors are identified using both simple and adjusted logistic regression (LR) models. In this review, common risk factors include antimicrobial resistance (AMR) patterns, electronic health records, prior antibiotic use, history of AMR conditions, or bacterial colonization. These factors are strongly correlated with AMR and are frequently used as predictors in machine learning (ML) models and risk score evaluations [39]. However, determining whether additional factors, such as underlying health conditions, can improve prediction accuracy remains difficult. Established variables, like proton pump inhibitor (PPI) usage [41], may sometimes be overlooked, especially in retrospective studies. Therefore, more prospective studies are required to better understand the effect of PPI use.

Another method for refining predictors is through the application of feature selection algorithms [42]. While some predictors identified by these algorithms are consistent with those proposed by LR models or prior research, others, such as the date of admission, may have unclear associations with AMR. It is recommended that both domain expertise and a systematic approach be employed to effectively process the large volumes of data derived from healthcare systems [39].

The results of this systematic review indicate that machine learning (ML) prediction models could support antibiotic prescribing decisions for bacterial infections caused by carbapenem-resistant pathogens. These findings are consistent with those observed in a previous review [39]. However, other studies have examined the comparative effectiveness of ML algorithms versus traditional risk scores, with mixed results [27,32]. The evidence in this area remains inconsistent. For example, one systematic review assessing diagnostic or prognostic models for binary outcomes based on clinical data found no substantial evidence that ML outperforms logistic regression (LR) [43], contrary to the conclusions of two other systematic reviews [44,45]. Beunza et al. [44] suggested that ML can improve the diagnostic and prognostic performance of conventional regression methods, while Sufriyana et al. [45] advocated for reevaluating existing LR models and comparing them to algorithms that follow standardized protocols. Although risk scores may provide useful decision-making support at the bedside, it is believed that integrating health information systems with ML algorithms can leverage large datasets to address this challenge more effectively [39]. The key advantage of ML lies in its ability to continuously improve through learning, leading to enhanced model accuracy and diverse applications in healthcare. Unlike traditional statistical methods, ML does not depend on fulfilling specific assumptions, which are often not met or assessed in medical research [46]. As such, the choice of algorithm should be guided by the specific research question and the context of its application.

Incomplete data are inevitable in retrospective cohort studies, leading to statistical complexity and bias in ML predictions. Another challenge is data imbalance in the AMR prediction model [39]. This imbalance negatively affects prediction performance, as classifiers tend to favor the majority class to minimize overall error rates [47]. A similar issue of data imbalance has been highlighted in other domains involving ML applications, such as blocking bug prediction models. The systematic review by Brown et al. [48] underscores how imbalanced datasets can lead to biased evaluation metrics, such as accuracy, and emphasizes the need for more robust approaches to validate prediction models effectively. To mitigate this issue, techniques such as resampling, adjusting hyperparameters, and carefully selecting methods may be employed [39]. Future researchers should collaborate with diverse teams to develop high-quality models.

The developed model still has a long way to go before it can be implemented in clinical practice. While these models offer accurate predictions, making decisions in daily medical practice is complex. In Oonsivilai et al.'s study [49], the final antibiotic selection was based on predicted AMR results and cost, with an optimal threshold of 0.21 established to avoid one necessary carbapenem. Similarly, in Stracy et al.'s study [50], antibiotic prescription supported by ML was useful in minimizing post-treatment-acquired re-resistance. However, various factors such as patient preference, economic status, and medical service availability influence the final treatment option. Therefore, while good prediction performance can serve as a surrogate outcome, it cannot be the sole determining factor. Some randomized controlled trials have explored the effect of ML intervention on patient prognosis [51,52], but there has been no investigation into AMR prediction models. Clinical practitioners are more interested in decreased AMR-attributable mortality by ML assistance rather than predictive accuracy [39]. A well-developed ML model needs external validation in another dataset and evaluation of endpoint outcomes in clinical trials or real-world studies before it can be integrated into daily practice.

At present, there is no universally accepted instrument for evaluating the risk of bias in machine learning (ML) prediction studies. Delpino et al. [53] employed the TRIPOD statement [54] to assess the quality of studies, while Fleuren et al. [55] and Christodoulou et al. [43] utilized the QUADAS-2 criteria [56]. The TRIPOD statement is primarily used as a checklist rather than a dedicated bias evaluation tool, while the QUADAS-2 criteria are commonly applied to gauge the quality of diagnostic accuracy studies [56]. Similarly to the current review, the PROBAST tool [21] was also applied in a previous study to assess ML methods for predicting antimicrobial resistance [39].

This study had several limitations. High heterogeneity was observed among the analyzed studies due to differences in outcomes, predictors, ML algorithms, hyperparameters, and populations. Most included studies were assessed as having a high risk of bias, which makes it challenging to perform a meta-analysis based on varying levels of bias risk. Two systematic reviews of predictive models using ML also indicated high heterogeneity among the studies included in their analyses [39,43]. For example, the systematic review of ML in predicting AMR revealed a heterogeneity greater than 97% [39]. Another assessment found that for 145 comparisons with a low risk of bias, there was no difference in AuROC between LR and ML (0.00, 95% CI −0.18 to 0.18). However, in 137 comparisons with a high likelihood of bias, ML had an AuROC 0.34 (0.20–0.47) higher [43]. It is important to highlight that Cochrane suggests that if the $I^2$ statistic is above 50%, substantial heterogeneity may be present, and caution should be taken when interpreting the results [57,58].

The high heterogeneity observed in this study raises concerns about the generalizability of the results. Future studies should aim to mitigate the impact of heterogeneity by employing strategies such as meta-regression or subgroup analyses to identify and address sources of variability. These methods, although not applied in the current review, could provide deeper insights into the influence of specific factors, such as study design, population characteristics, and ML algorithm choice, on outcomes.

Additionally, while digital methods such as machine learning have shown great promise in various medical applications, they rely heavily on access to computational resources and stable internet connectivity. This limitation is particularly significant in resource-limited settings, such as developing countries, where the burden of antimicrobial resistance and associated deaths is highest [3,4]. Future research should explore ways to adapt ML tools for offline use, develop lightweight algorithms that can operate on low-resource devices, or integrate these tools with existing healthcare systems in such regions.

Lastly, this study highlights the need for more actionable insights in future research. Specifically, efforts should focus on developing and adopting standardized reporting guidelines for

ML studies in AMR research to reduce variability, ensure methodological transparency, and facilitate reproducibility. Additionally, improving data quality and accessibility, and testing the scalability of ML models in real-world healthcare settings, are critical steps. Addressing these areas will enhance the practical applicability of ML in tackling AMR and other global health challenges.

In conclusion, this systematic review was conducted on the prediction of AMR of CHPP using ML and considering antimicrobial susceptibility tests in a real-world context. It was found that this approach achieved satisfactory results in most of the included studies. Therefore, ML prediction could be a promising technology for assisting in antibiotic selection. However, it is important to introduce a recognized guideline into this field to ensure consistency in future studies, and these prediction models should also be evaluated in randomized controlled trials.

## Supporting information

**S1 File. S1 PRISMA 2020 checklist.**
(DOCX)

**S1 Table. List of excluded studies with reasons.**
(DOCX)

## Author contributions

**Conceptualization:** Carlos M Ardila.

**Data curation:** Carlos M Ardila.

**Formal analysis:** Carlos M Ardila, Daniel González-Arroyave.

**Investigation:** Carlos M Ardila, Daniel González-Arroyave.

**Methodology:** Carlos M Ardila, Daniel González-Arroyave, Sergio Tobón.

**Project administration:** Carlos M Ardila.

**Supervision:** Carlos M Ardila, Sergio Tobón.

**Validation:** Carlos M Ardila, Daniel González-Arroyave, Sergio Tobón.

**Visualization:** Carlos M Ardila, Daniel González-Arroyave, Sergio Tobón.

**Writing – original draft:** Carlos M Ardila, Daniel González-Arroyave.

**Writing – review & editing:** Carlos M Ardila, Daniel González-Arroyave, Sergio Tobón.

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
