## [Decision Letter · Decision Letter 0]

13 Dec 2024

PONE-D-24-19315Machine learning for predicting antimicrobial resistance in critical and high-priority pathogens: A systematic review considering antimicrobial susceptibility tests in real-world healthcare settings.PLOS ONE

Dear Dr. Ardila,

Thank you for submitting your manuscript to PLOS ONE. After careful consideration, we feel that it has merit but does not fully meet PLOS ONE’s publication criteria as it currently stands. Therefore, we invite you to submit a revised version of the manuscript that addresses the points raised during the review process.

We look forward to receiving your revised manuscript.

Kind regards,

Mohamed O Ahmed, Ph.D

Academic Editor

PLOS ONE

Journal Requirements:

3. As required by our policy on Data Availability, please ensure your manuscript or supplementary information includes the following:

Reviewers' comments:

Reviewer's Responses to Questions

**Comments to the Author**

1. Is the manuscript technically sound, and do the data support the conclusions?

Reviewer #1: Partly

Reviewer #2: Yes

Reviewer #3: Yes

2. Has the statistical analysis been performed appropriately and rigorously? 

Reviewer #1: N/A

Reviewer #2: Yes

Reviewer #3: Yes

3. Have the authors made all data underlying the findings in their manuscript fully available?

Reviewer #1: No

Reviewer #2: Yes

Reviewer #3: Yes

4. Is the manuscript presented in an intelligible fashion and written in standard English?

Reviewer #1: Yes

Reviewer #2: Yes

Reviewer #3: Yes

5. Review Comments to the Author

Reviewer #1: This is a well written report which provides a valuable contribution to avenues relevant to the exploration of AMR and AMS. Some sections require further clarification to provide context to the approach undertaken.

Reviewer #2: 1. This manuscript heavily discuses on AUROC. It would be helpful if the authors provided a figure that summarizes AUROC, and their conclusions based on the various articles they studied.

2. Digital methods, such as machine learning, are undoubtedly valuable in various medical applications, as demonstrated in this study. However, these methods rely on the availability of computers and stable internet, which are often lacking in remote areas. In addition to outlining the benefits of this method, the authors should also address its limitations, particularly in areas with limited resources, such as developing countries. This is particularly important considering the author's statement that most deaths due to ARBs occur in developing countries (Line 60-61).

3. Line 724: Is "metho" referring to a method?

Reviewer #3: The manuscript is well-structured and tackles an important topic. However, It could be improved further by tackling the following:

1. Adding more actionable insights or suggestions for future research would further strengthen its contribution to the field

2. You acknowledge the high heterogeneity across the included studies, which is appropriate. However, I recommend elaborating on how this heterogeneity might affect the generalizability of the results. Discussing possible strategies for reducing the impact of heterogeneity (e.g., meta-regression or subgroup analysis) would be useful, even if they weren’t applied in the current review.

3. Consider grouping the results by key ML models or outcomes (e.g., by pathogens or prediction performance). This will help readers draw clearer connections between different studies

6. PLOS authors have the option to publish the peer review history of their article (what does this mean? ). If published, this will include your full peer review and any attached files.

**Do you want your identity to be public for this peer review?** For information about this choice, including consent withdrawal, please see our Privacy Policy .

Reviewer #1: No

Reviewer #2: **Yes: ** Retno Wahyuningsih

Reviewer #3: No

---

## [Author Response · Author response to Decision Letter 1]

18 Dec 2024

Dear Reviewers,

We are grateful for the constructive comments you provided, which helped us to improve the manuscript significantly.

Our responses to your comments are outlined below and highlighted in green in the new version.

Reviewer #1: This is a well written report which provides a valuable contribution to avenues relevant to the exploration of AMR and AMS. Some sections require further clarification to provide context to the approach undertaken.

1. Line 56. Consider referencing the primary source rather than a secondary source.

Response: The primary resource was referenced following the recommendations.

2. Line 210. Aspects of data extraction is presented, however ‘personalised data extraction methods’ needs to be better explained.

Response: We thank the reviewer for this valuable comment. To address this, we have clarified the nature of the personalized data extraction methods used in our study. Specifically, we have detailed the development and use of customized and study-specific data extraction templates tailored to our research objectives. These templates facilitated the systematic collection of essential variables and publication-specific details, ensuring consistency and reliability across independent data collection efforts. The revised text can be found in Section 2.6 (Data Collection).

3. Line 228 to 231. A comprehensive narrative of the data analysis and reporting approach is needed.

Response: We appreciate the reviewer’s insightful suggestion. In response, we have expanded the description of our data analysis and reporting approach in Section 2.8 (Summary Measurements). Specifically, we have detailed the use of descriptive statistics for summarizing continuous outcomes and provided additional information on how we assessed heterogeneity to determine the feasibility of meta-analysis. Furthermore, we have clarified that in the absence of sufficient homogeneity, a qualitative synthesis was conducted to provide a narrative summary of the findings. This revision offers a more comprehensive explanation of our approach and addresses the reviewer's concern.

4. Line 244. Figure 1: A more detailed PRISMA flow chart is needed.

Response: We thank the reviewer for highlighting this point. In response, we have revised Figure 1 to include additional details for each stage of the study selection process. Considering that the study was submitted several months ago, we updated our search. We have detailed the records obtained in each database and provided clear descriptions of the reasons for exclusions during screening and full-text eligibility assessments, as well as the criteria used at each step. This enhanced figure offers a more transparent and detailed representation of the study selection methodology.

Reviewer #2:

1. This manuscript heavily discuses on AUROC. It would be helpful if the authors provided a figure that summarizes AUROC, and their conclusions based on the various articles they studied.

Response: A figure was provided following the recommendations.

Fig. 2. Heatmap of Area Under the Receiver Operating Characteristic Curve (AUROC) Values for Various Machine Learning Models Across Studies. This figure presents a heatmap summarizing the AUROC values reported in different studies for various machine learning models. Each cell represents the AUROC value reported in the corresponding study for the specific model, with darker colors indicating higher AUROC values (closer to 1.0), signifying better model performance. Blank cells indicate cases where no AUROC value was reported. Studies are arranged along the vertical axis, while machine learning models are organized along the horizontal axis. This heatmap provides a visual comparison of model performance across studies and highlights models that consistently achieve high AUROC scores.

2. Digital methods, such as machine learning, are undoubtedly valuable in various medical applications, as demonstrated in this study. However, these methods rely on the availability of computers and stable internet, which are often lacking in remote areas. In addition to outlining the benefits of this method, the authors should also address its limitations, particularly in areas with limited resources, such as developing countries. This is particularly important considering the author's statement that most deaths due to ARBs occur in developing countries (Line 60-61).

Response: The suggestion was recognized and the following text was added to the study limitations: Additionally, while digital methods such as machine learning have shown great promise in various medical applications, they rely heavily on access to computational resources and stable internet connectivity. This limitation is particularly significant in resource-limited settings, such as developing countries, where the burden of antimicrobial resistance and associated deaths is highest [3,4]. Future research should explore ways to adapt ML tools for offline use, develop lightweight algorithms that can operate on low-resource devices, or integrate these tools with existing healthcare systems in such regions.

3. Line 724: Is "metho" referring to a method?

Response: The typo was resolved.

Reviewer #3: The manuscript is well-structured and tackles an important topic. However, it could be improved further by tackling the following:

1. Adding more actionable insights or suggestions for future research would further strengthen its contribution to the field.

Response: The suggestion was recognized and the following text was added to the study limitations: Future studies should aim to mitigate the impact of heterogeneity by employing strategies such as meta-regression or subgroup analyses to identify and address sources of variability. These methods, although not applied in the current review, could provide deeper insights into the influence of specific factors, such as study design, population characteristics, and ML algorithm choice, on outcomes.

Additionally, while digital methods such as machine learning have shown great promise in various medical applications, they rely heavily on access to computational resources and stable internet connectivity. This limitation is particularly significant in resource-limited settings, such as developing countries, where the burden of antimicrobial resistance and associated deaths is highest [3,4]. Future research should explore ways to adapt ML tools for offline use, develop lightweight algorithms that can operate on low-resource devices, or integrate these tools with existing healthcare systems in such regions.

Lastly, the study highlights the need for more actionable insights in future research. Specifically, efforts should focus on developing standardized reporting guidelines for ML studies to reduce variability, improving data quality and accessibility, and testing the scalability of ML models in real-world healthcare settings. Addressing these areas will enhance the practical applicability of ML in tackling AMR and other global health challenges.

2. You acknowledge the high heterogeneity across the included studies, which is appropriate. However, I recommend elaborating on how this heterogeneity might affect the generalizability of the results. Discussing possible strategies for reducing the impact of heterogeneity (e.g., meta-regression or subgroup analysis) would be useful, even if they weren’t applied in the current review.

Response: The suggestion was recognized and the following text was added to the study limitations:

The high heterogeneity observed in this study raises concerns about the generalizability of the results. Future studies should aim to mitigate the impact of heterogeneity by employing strategies such as meta-regression or subgroup analyses to identify and address sources of variability. These methods, although not applied in the current review, could provide deeper insights into the influence of specific factors, such as study design, population characteristics, and ML algorithm choice, on outcomes.

Additionally, while digital methods such as machine learning have shown great promise in various medical applications, they rely heavily on access to computational resources and stable internet connectivity. This limitation is particularly significant in resource-limited settings, such as developing countries, where the burden of antimicrobial resistance and associated deaths is highest [3,4]. Future research should explore ways to adapt ML tools for offline use, develop lightweight algorithms that can operate on low-resource devices, or integrate these tools with existing healthcare systems in such regions.

Lastly, the study highlights the need for more actionable insights in future research. Specifically, efforts should focus on developing standardized reporting guidelines for ML studies to reduce variability, improving data quality and accessibility, and testing the scalability of ML models in real-world healthcare settings. Addressing these areas will enhance the practical applicability of ML in tackling AMR and other global health challenges.

3. Consider grouping the results by key ML models or outcomes (e.g., by pathogens or prediction performance). This will help readers draw clearer connections between different studies.

Response: A figure was provided following the recommendations.

Fig. 2. Heatmap of Area Under the Receiver Operating Characteristic Curve (AUROC) Values for Various Machine Learning Models Across Studies. This figure presents a heatmap summarizing the AUROC values reported in different studies for various machine learning models. Each cell represents the AUROC value reported in the corresponding study for the specific model, with darker colors indicating higher AUROC values (closer to 1.0), signifying better model performance. Blank cells indicate cases where no AUROC value was reported. Studies are arranged along the vertical axis, while machine learning models are organized along the horizontal axis. This heatmap provides a visual comparison of model performance across studies and highlights models that consistently achieve high AUROC scores.

---

## [Decision Letter · Decision Letter 1]

7 Jan 2025

PONE-D-24-19315R1Machine learning for predicting antimicrobial resistance in critical and high-priority pathogens: A systematic review considering antimicrobial susceptibility tests in real-world healthcare settings.PLOS ONE

Dear Dr. Ardila,

Thank you for submitting your manuscript to PLOS ONE. After careful consideration, we feel that it has merit but does not fully meet PLOS ONE’s publication criteria as it currently stands. Therefore, we invite you to submit a revised version of the manuscript that addresses the points raised during the review process.

**ACADEMIC EDITOR: ***Comments from PLOS Editorial Office:* We note that one or more reviewers has recommended that you cite specific previously published works. As always, we recommend that you please review and evaluate the requested works to determine whether they are relevant and should be cited. It is not a requirement to cite these works. We appreciate your attention to this request.

We look forward to receiving your revised manuscript.

Kind regards,

Mohamed O Ahmed, Ph.D

Academic Editor

PLOS ONE

Journal Requirements:

Reviewers' comments:

Reviewer's Responses to Questions

**Comments to the Author**

1. If the authors have adequately addressed your comments raised in a previous round of review and you feel that this manuscript is now acceptable for publication, you may indicate that here to bypass the “Comments to the Author” section, enter your conflict of interest statement in the “Confidential to Editor” section, and submit your "Accept" recommendation.

Reviewer #2: All comments have been addressed

Reviewer #3: All comments have been addressed

2. Is the manuscript technically sound, and do the data support the conclusions?

Reviewer #2: Yes

Reviewer #3: Partly

3. Has the statistical analysis been performed appropriately and rigorously? 

Reviewer #2: Yes

Reviewer #3: N/A

4. Have the authors made all data underlying the findings in their manuscript fully available?

Reviewer #2: (No Response)

Reviewer #3: Yes

5. Is the manuscript presented in an intelligible fashion and written in standard English?

Reviewer #2: Yes

Reviewer #3: No

6. Review Comments to the Author

Reviewer #2: (No Response)

Reviewer #3: 1. The introduction presents interesting statistics on the estimated number of infections and deaths, amongst others. It would be helpful if the authors stated clearly in each instance of the statistics whether it is global or related to developing countries.

2. Authors should reference and discuss previous systematic literature reviews in the domain and make a case for the relevance of the current review. Below are some examples of related reviews in the domain:

a) Lv, G., & Wang, Y. (2024). Machine learning-based antibiotic resistance prediction models: An updated systematic review and meta-analysis. Technology and Health Care, (Preprint), 1-18.

b) Sakagianni, A., Koufopoulou, C., Feretzakis, G., Kalles, D., Verykios, V. S., & Myrianthefs, P. (2023). Using machine learning to predict antimicrobial resistance―a literature review. Antibiotics, 12(3), 452.

c) E. Elyan, A. Hussain, A. Sheikh, A. A. Elmanama, P. Vuttipittayamongkol and K. Hijazi, "Antimicrobial Resistance and Machine Learning: Challenges and Opportunities," in IEEE Access, vol. 10, pp. 31561-31577, 2022, doi: 10.1109/ACCESS.2022.3160213.

3. The author identifies data imbalance as one of the issues with existing data in the domain. Authors can, therefore, consider referencing the following article:

Brown, S. A., Weyori, B. A., Adekoya, A. F., Kudjo, P. K., & Mensah, S. (2022). Predicting Blocking Bugs with Machine Learning Techniques: A Systematic Review. International Journal of Advanced Computer Science and Applications, 13(6).

4. Authors should suggest standardizing ML study reporting in AMR research, given the gaps identified in this work.

5. Authors should proofread their work thoroughly to correct grammatical errors.

7. PLOS authors have the option to publish the peer review history of their article (what does this mean? ). If published, this will include your full peer review and any attached files.

**Do you want your identity to be public for this peer review?** For information about this choice, including consent withdrawal, please see our Privacy Policy .

Reviewer #2: **Yes: ** Retno Wahyuningsih

Reviewer #3: **Yes: ** Selasie Aformaley Brown,Ph.D

---

## [Author Response · Author response to Decision Letter 2]

10 Jan 2025

Dear Reviewer,

Our responses to your comments are outlined below and highlighted in green in the new version.

Reviewer #3:

1. The introduction presents interesting statistics on the estimated number of infections and deaths, amongst others. It would be helpful if the authors stated clearly in each instance of the statistics whether it is global or related to developing countries.

Response: We have carefully reviewed the introduction and incorporated the suggested clarifications. In the revised version of the manuscript, we have explicitly stated that the statistics cited are based on global data. For example:

The projected number of deaths from bacterial infections by 2050 is described as occurring on a global scale.

The data from 2019 on deaths linked to antibiotic-resistant bacteria (ARBs) is specified as worldwide.

The statistics on methicillin-resistant S. aureus (MRSA) and other multidrug-resistant pathogens are clarified as global or occurring across the globe.

2. Authors should reference and discuss previous systematic literature reviews in the domain and make a case for the relevance of the current review. Below are some examples of related reviews in the domain:

a) Lv, G., & Wang, Y. (2024). Machine learning-based antibiotic resistance prediction models: An updated systematic review and meta-analysis. Technology and Health Care, (Preprint), 1-18.

b) Sakagianni, A., Koufopoulou, C., Feretzakis, G., Kalles, D., Verykios, V. S., & Myrianthefs, P. (2023). Using machine learning to predict antimicrobial resistance―a literature review. Antibiotics, 12(3), 452.

c) E. Elyan, A. Hussain, A. Sheikh, A. A. Elmanama, P. Vuttipittayamongkol and K. Hijazi, "Antimicrobial Resistance and Machine Learning: Challenges and Opportunities," in IEEE Access, vol. 10, pp. 31561-31577, 2022, doi: 10.1109/ACCESS.2022.3160213.

Response: Thank you for your thoughtful suggestion to include additional systematic literature reviews in our manuscript to further support the relevance of our current review. We appreciate the references provided and acknowledge their contributions to the field. However, we would like to respectfully clarify that our systematic review already incorporates and discusses several relevant and high-impact systematic reviews and meta-analyses addressing antimicrobial resistance (AMR) and machine learning-based prediction models. These include:

- O’Neill J (2016). Tackling Drug-Resistant Infections Globally.

- Antimicrobial Resistance Collaborators. Global burden of bacterial antimicrobial resistance in 2019: a systematic analysis. Lancet. 2022;399(10325):629-655. doi: 10.1016/S0140-6736(21)02724-0

- Tang et al. (2022) – A systematic review and meta-analysis on machine learning in predicting antimicrobial resistance [39].

- Pormohammad et al. (2019) – A systematic review and meta-analysis on antibiotic resistance in E. coli strains isolated from multiple sources [6].

- Christodoulou et al. (2019) – A systematic review evaluating machine learning versus logistic regression for clinical prediction models [43].

- Beunza et al. (2019) – A comparison of machine learning algorithms for predicting clinical events, [44].

- Sufriyana et al. (2020) – A systematic review and meta-analysis comparing multivariable logistic regression with machine learning algorithms [45].

- Delpino et al. (2022) – A systematic review on machine learning applications for predicting chronic diseases [52].

Additionally, we have included foundational reviews such as O’Neill (2016) [2] and the Antimicrobial Resistance Collaborators (2022) [3], which provide a broader context of AMR's global impact and underscore the urgency of predictive approaches.

Given the comprehensive nature of these cited works, we believe our manuscript already provides a well-rounded background without overwhelming the reader with excessive references.

3. The author identifies data imbalance as one of the issues with existing data in the domain. Authors can, therefore, consider referencing the following article:

Brown, S. A., Weyori, B. A., Adekoya, A. F., Kudjo, P. K., & Mensah, S. (2022). Predicting Blocking Bugs with Machine Learning Techniques: A Systematic Review. International Journal of Advanced Computer Science and Applications, 13(6).

Response: We appreciate your insightful suggestion to include the recommended reference by Brown et al. (2022) to strengthen our discussion on data imbalance challenges in machine learning applications. In response, we have incorporated the reference into the discussion section of our revised manuscript. Specifically, we highlight how the study by Brown et al. underscores the impact of imbalanced datasets on evaluation metrics and emphasizes the need for more robust validation approaches. This addition provides a broader perspective on addressing data imbalance issues across domains involving machine learning models.

4. Authors should suggest standardizing ML study reporting in AMR research, given the gaps identified in this work.

Response: We have revised the discussion section to explicitly emphasize the importance of developing and adopting standardized reporting guidelines for ML studies. This addition highlights the necessity of reducing variability, ensuring methodological transparency, and facilitating reproducibility in future AMR research.

5. Authors should proofread their work thoroughly to correct grammatical errors.

Response: A thorough review of the manuscript was conducted, and grammatical errors were corrected.

---

## [Editor Report · Decision Letter 2]

4 Feb 2025

Machine learning for predicting antimicrobial resistance in critical and high-priority pathogens: A systematic review considering antimicrobial susceptibility tests in real-world healthcare settings.

PONE-D-24-19315R2

Dear Dr. Ardila,

We’re pleased to inform you that your manuscript has been judged scientifically suitable for publication and will be formally accepted for publication once it meets all outstanding technical requirements.

Kind regards,

Mohamed O Ahmed, Ph.D

Academic Editor

PLOS ONE

---

## [Editor Report · Acceptance letter]

PONE-D-24-19315R2

PLOS ONE

Dear Dr. Ardila,

I'm pleased to inform you that your manuscript has been deemed suitable for publication in PLOS ONE. Congratulations! Your manuscript is now being handed over to our production team.

Kind regards,

on behalf of

Dr. Mohamed O Ahmed

Academic Editor

PLOS ONE